# LSSmScarlet2 and LSSmScarlet3, Chemically Stable Genetically Encoded Red Fluorescent Proteins with a Large Stokes’ Shift

**DOI:** 10.3390/ijms231911051

**Published:** 2022-09-21

**Authors:** Oksana M. Subach, Anna V. Vlaskina, Yulia K. Agapova, Kiryl D. Piatkevich, Maxim V. Patrushev, Valeriya R. Samygina, Fedor V. Subach

**Affiliations:** 1Complex of NBICS Technologies, National Research Center “Kurchatov Institute”, 123182 Moscow, Russia; 2School of Life Sciences, Westlake University, Hangzhou 310024, China; 3Westlake Laboratory of Life Sciences and Biomedicine, Hangzhou 310024, China; 4Institute of Basic Medical Sciences, Westlake Institute for Advanced Study, Hangzhou 310024, China; 5Institute of Crystallography of Federal Research Scientific Center “Crystallography and Photonics” of the Russian Academy of Sciences, 119333 Moscow, Russia

**Keywords:** genetically encoded red fluorescent proteins, protein engineering, fluorescence imaging, large Stokes’ shift, LSSmScarlet2, LSSmScarlet3, red fluorescent, fluorescent protein

## Abstract

Red fluorescent proteins with a large Stokes’ shift (LSSRFPs) are genetically encoded and efficiently excited by 488 nm light, allowing simultaneous dual-color one- and two-photon fluorescence imaging and fluorescence correlation spectroscopy in combination with green fluorescent proteins FPs. Recently, based on the conventional bright mScarlet RFP, we developed the LSSRFP LSSmScarlet. LSSmScarlet is characterized by two pKa values at pH values of 1.9 and 5.8. In this study, we developed improved versions of LSSmScarlet, named LSSmScarlet2 and LSSmScarlet3, which are characterized by a Stokes’ shift of 128 nm and extreme pH stability with a single pKa value of 2.2. LSSmScarlet2 and LSSmScarlet3 had 1.8-fold faster and 3-fold slower maturation than LSSmScarlet, respectively. In addition, both LSSRFPs were 1.5- to 1.6-fold more photostable and more chemically resistant to denaturation by guanidinium chloride and guanidinium thiocyanate. We also compared the susceptibility of the LSSmScarlet2, LSSmScarlet3, and other LSSRFPs to the reagents used for whole-mount imaging, expansion microscopy, and immunostaining techniques. Due to higher pH stability and faster maturation, the LSSmScarlet3-LAMP3 fusion was 2.2-fold brighter than LSSmScarlet-LAMP3 in lysosomes of mammalian cells. The LSSmScarlet3-hLAMP2A fusion was similar in brightness to LSSmScarlet-hLAMP2A in lysosomes. We successfully applied the monomeric LSSmScarlet2 and LSSmScarlet3 proteins for confocal imaging of structural proteins in live mammalian cells. We also solved the X-ray structure of the LSSmScarlet2 protein at a resolution of 1.41 Å. Site-directed mutagenesis of the LSSmScarlet2 protein demonstrated the key role of the T74 residue in improving the pH and chemical stability of the LSSmScarlet2 protein.

## 1. Introduction

Red fluorescent proteins with a large Stokes’ shift (LSSRFPs) are genetically encoded and have a large difference (approximately 100 nm) between the excitation and emission peaks. They efficiently absorb blue light and emit red fluorescence. The LSSRFPs provide separate colors for multicolor fluorescence microscopy and can be utilized for simultaneous imaging of several activities in living cells [1]. LSSRFPs can be simultaneously visualized in conjunction with green FPs and biosensors under common confocal and two-photon microscopes with single-wavelength excitation [2].

There is a great demand for chemically stable FPs for application in modern microscopy methods, such as superresolution (SR) imaging [3], expansion microscopy (ExM) [4], and correlative light and electron microscopy (CLEM) [5]. In many microscopy methods, biological samples should be chemically fixed with paraformaldehyde (PFA) or glutaraldehyde (GA) solutions. FPs can be adversely affected by aldehyde fixation. Recently, several chemically stable green FPs, including mGreenLantern [6] and green-to-red photoconvertible FPs, mEos4a and mEos4b [5], have been reported.

In addition to chemical stability, FPs should have favorable properties for broad utility, including high pH stability, brightness, photostability, fast and complete maturation, and a monomeric state. The most pH-stable FPs to date are mTagBFP2 (pKa 2.7) [7] and mCRISPRed (pKa 2.14) [8]. However, mCRISPRed is less stable at alkaline pH than at lower pH values since its fluorescence decreases at pH values higher than 7 with a pKa of 9.8. LSSmScarlet is stable at alkaline pH, but under acidic conditions, it has two pH transitions with pKa values of 1.91 and 5.78 [9]. It is known that the pH in different organelles inside mammalian cells may vary from 4.5 to 8.0 [10]. It would be beneficial for modern microscopy techniques as well as applications in acidic environments, such as lysosomes, and in slightly alkaline environments, such as mitochondria, to develop LSSRFPs with pH stability throughout the entire range of physiologically relevant pH values.

In this study, using the previously reported LSSmScarlet LSSRFP as a template, we developed pH-, photo-, and chemically stable LSSRFPs, named LSSmScarlet2 and LSSmScarlet3. LSSmScarlet2 and LSSmScarlet3 sustain their fluorescence in the range of pH from 2.5 to 10, although the molecular brightness of LSSmScarlet2 and LSSmScarlet3 is 1.3- to 1.5-fold lower than that of the parental LSSmScarlet. Both LSSmScarlet2 and LSSmScarlet3 are 1.5- to 1.6-fold more photostable and more chemically stable than LSSmScarlet. The faster maturation and higher pH stability in acidic conditions of LSSmScarlet3 compared to LSSmScarlet result in a 2.2-fold higher brightness of LSSmScarlet3 in lysosomes in fusion with lysosomal associated membrane protein 3 (LAMP3). The monomeric properties of LSSmScarlet2 and LSSmScarlet3 ensure their correct localization in fusions with cytoskeletal proteins in mammalian cells. To investigate the mechanisms of the pH and chemical stability, we determined the crystal structure of LSSmScarlet2 at a 1.4 Å resolution. Finally, we carried out site-directed mutagenesis of LSSmScarlet2 and revealed the impact of each mutation on the pH and chemical stability of LSSmScarlet2 compared to LSSmScarlet.

## 2. Results and Discussion

### 2.1. Developing the Large Stokes’ Shift Fluorescent Protein LSSmScarlet2 in E. coli

To develop enhanced LSSRFPs, LSSmScarlet and its pH-stable mutant LSSmScarlet/R198H were subjected to six rounds of directed molecular evolution using error-prone PCR. We chose LSSmScarlet/R198H as a template for mutagenesis due to its high pH stability with a pKa value of 2.4 [9]. During each round of evolution, we screened for mutants exhibiting the highest pH stability when expressed in *E. coli* cells. First, we selected *E. coli* colonies on Petri dishes that possessed the highest fluorescence in the LSSRed channel (excitation and emission at 480/40 and 620/60 nm, respectively) 1 h after spraying 100 mM NaOAc (pH 1.0) buffer. Next, the selected clones were expressed and analyzed in bacterial lysates using a fluorescence plate reader. We chose 1–2 clones based on the ratio of their fluorescence at pH 7.5 to their fluorescence at pH 3.5, which was closer to 1 for the better clones. After six rounds of directed molecular evolution, we selected the final variant LSSmScarlet/I74T/D170G/D177H/Q189L/Y215F/G226D, named LSSmScarlet2, which had the highest pH stability and molecular brightness at 488 nm excitation. LSSmScarlet2 had six mutations relative to the LSSmScarlet template; five of them were located outside of the β-barrel (Figure 1).

### 2.2. In Vitro Characterization of the Purified LSSmScarlet2 Protein

First, we characterized the spectral properties and molecular brightness of the developed LSSmScarlet2 LSSRFP (Figure 2a and Table 1); the properties of LSSmScarlet 3 are described in Section 2.5. LSSmScarlet2 had absorption/excitation/emission peaks at 467/470/600 nm, respectively (Figure 2a and Table 1). LSSmScarlet2 had a Stokes’ shift of 128 nm (Table 1). The molecular brightness at the maximum of excitation and at 488 nm for the developed LSSmScarlet2 was 1.5- and 1.45-fold dimmer than that for LSSmScarlet (Table 1).

As pH varies across cellular organelles inside cells from 4.5 in lysosomes to 8.0 in mitochondria [10], pH stability of FPs is an important property for live cell imaging. Therefore, we assessed the dependence of LSSmScarlet2 fluorescence on pH (Figure 2c and Table 1). LSSmScarlet2 demonstrated maximal fluorescence in the pH range from 2.5 to 10.0 with a pKa value of 2.19. In contrast, the LSSmScarlet fluorescence showed two steps of decrease during acidification: a drop of approximately 2-fold with a pKa value of 5.78 and further complete quenching of fluorescence with a pKa value of 1.91 (Figure 2c and Table 1). Thus, among the most pH-stable LSSRFPs, such as the mCRISPRed and LSSmScarlet proteins, LSSmScarlet2 revealed the highest pH stability at both acidic and alkaline pH values.

We next assessed the maturation rate for LSSmScarlet2 at 37 °C using protein extracted from bacterial cells (Figure 2d and Table 1). The maturation half-time for LSSmScarlet2 was 3-fold longer than that for the original LSSmScarlet. The slow maturation rate of LSSmScarlet2 prompted us to search for another faster-maturing LSSmScarlet variant.

### 2.3. Structural Characterization of the LSSmScarlet2 Protein

To explain the high pH stability of the LSSmScarlet2 protein and to elucidate the effects of mutations introduced into LSSmScarlet during its directed molecular evolution, we determined the crystal structure of LSSmScarlet2 using X-ray diffraction at 1.41 Å resolution (Figure 3 and Table 2). LSSmScarlet and LSSmScarlet2 have almost identical structures. We observed the LSSmScarlet2 protein as a monomer in the crystal at pH 6.0 according to estimation by the PISA server [11] in a fluorescent state (Figure 3a). LSSmScarlet2 has a canonical β-barrel overall structure with the DsRed-like chromophore, which consists of ^69^MYG^71^ amino acids and is located in the middle of the central α−helix inside the β-barrel (Figure 3a,b). The MYG-chromophore in LSSmScarlet2 makes five direct hydrogen bonds (H-bonds) to residues R71, W94, R96, D147, and E216 and additional H-bonds through the water molecules to residues T112, E145, and L200 (Figure 3c,d). In the LSSmScarlet2 structure, two water-mediated contacts of the chromophore with Q65 and Q110 that were present in LSSmScarlet are absent. The phenolic hydroxyl group of the chromophore makes two H-bonds with the side chain of the D147 residue and with one internal water molecule. The negative charge of the pair D147/phenolic hydroxyl group of the chromophore is balanced by the positively charged R198 residue, which is close to the phenolic group of the chromophore and forms a solvent water-mediated H-bond with D147. Hence, the overall structure and chromophore surrounding the LSSmScarlet2 protein were similar to those of the LSSmScarlet protein.

The excited state proton transfer (ESPT) can explain a large Stokes’ shift in LSSmScarlet2 similar to the parental LSSmScarlet [9]. We observed a large Stokes’ shift in LSSmScarlet2, i.e., the excitation peak was shifted by 128 nm relative to the emission peak. LSSmScarlet had the same 128 nm large Stokes’ shift. Similar to LSSmScarlet, we attributed the large Stokes’ shift found in LSSmScarlet2 to ESPT from the phenolic hydroxyl group of the chromophore to the –COOH side chain group of the D147 residue. The large Stokes’ shifts observed in other reported LSSFPs were also explained by ESPT.

To explain the role of mutations found during LSSmScarlet2 development, we compared the chromophore’s environment for LSSmScarlet2 and the original LSSmScarlet protein (PDB: 7OIN) [9] (Figure 4). The chromophores of LSSmScarlet and LSSmScarlet2 matched perfectly. There were six mutations in LSSmScarlet2 compared to LSSmScarlet: I74T, D170G, D177H, Q189L, Y215F, and G226D (Figure 1). Only one mutation, I74T, is inside the β-barrel structure. I74T substitution results in the formation of a hydrogen bond with the S218 residue. The side chain of S218 in LSSmScarlet2 also has the second alternative conformation, in which the hydroxyl group forms a H-bond with the E216 residue. The remaining amino acid substitutions are far from the chromophore and lie on the surface of LSSmScarlet2. G170 is in the loop on the surface of LSSmScarlet2. D170 in LSSmScarlet forms a salt bridge with K140. H177 is in the single position in LSSmScarlet2 and forms a salt bridge with D161 and a stacking interaction with R179. In LSSmScarlet, the amino acid in position 177 is D177, which has two alternative conformations: one is similar to H177 in LSSmScarlet2, and the second contacts two H_2_O molecules. The Q189 side chain in LSSmScarlet forms a water-mediated hydrogen bond with OH Ser3, whereas the L189 side chain does not form any contacts. Substitution of Y215 in LSSmScarlet to F215 in LSSmScarlet2 resulted in loss of hydrogen bonding with the H_2_O molecule. In the LSSmScarlet2 crystal structure, the electron density for D226 is absent, but in the LSSmScarlet structure, the corresponding G226 residue is located at the C-terminal tail outside the β-barrel and should not influence LSSmScarlet2 fluorescence. Therefore, only the I74T mutation inside the β-barrel may affect the LSSmScarlet2 chromophore properties; the remaining amino acid changes are far away from the chromophore and likely influence the folding and maturation of LSSmScarlet2.

### 2.4. Site-Directed Mutagenesis of the LSSmScarlet2 LSSRFP

To elucidate the effects of amino acid mutations introduced in LSSmScarlet2 during its development on its pH stability, we generated six single-point mutants of LSSmScarlet2 containing T74I, G170D, H177D, L189Q, F215Y, or D226G substitutions and characterized their pH stability (Figure 5a). The LSSmScarlet2/T74I mutant had two pH transitions with pKa values of 2.03 and 6.16 (Figure 5a and Table 3). Similar to LSSmScarlet2, all other variants had one pH transition with pKa values varying from 1.88 to 2.33. For these five mutants, except for LSSmScarlet2/G170D, we observed a small drop in fluorescence above pH 4.5. Similar to LSSmScarlet2, the LSSmScarlet2/G170D mutant showed maximal fluorescence in the pH range from 2.5 to 10.0 (Figure 2c). Hence, the I74T mutation was the only one responsible for the improved pH stability of the LSSmScarlet2 protein, and among all LSSRFPs tested, the LSSmScarlet2/G170D mutant and LSSmScarlet2 demonstrated the highest pH stability in the entire pH range.

We next studied the impact of the introduced mutations on the chemical stability of the LSSmScarlet2 protein via incubation in guanidinium chloride (GuaHCl) for 24 h at room temperature (Figure 5b). The LSSmScarlet2/G170D mutant had chemical stability in guanidinium chloride (GuaHCl), similar to LSSmScarlet2 (Figure 5b and Table 3). The introduction of Ile at position 74 of LSSmScarlet2 (LSSmScarlet2/T74I mutant) led to a dramatic decrease in its chemical stability (Figure 5b and Table 3). Other mutations also influenced chemical stability but to a smaller extent. Therefore, the I74T mutation was the only mutation responsible for the improved chemical stability of the LSSmScarlet2 protein.

### 2.5. In Vitro Characterization of the Purified LSSmScarlet3 Protein

Since the LSSmScarlet2/G170D mutant (named LSSmScarlet3) had pH and chemical stabilities similar to those of LSSmScarlet2, its properties were characterized further. First, we measured the maturation rate for LSSmScarlet3 at 37 °C using protein extracted from bacteria (Figure 2d and Table 1). The maturation half-time values for LSSmScarlet3 were 1.8- and 5.4-fold faster than those for LSSmScarlet and LSSmScarlet2 LSSRFPs, respectively. Thus, in addition to the pH and chemical stability in guanidinium chloride described above in Section 2.4 (Table 3), LSSmScarlet3 demonstrated faster maturation than the LSSmScarlet and LSSmScarlet2 proteins.

Then, we estimated the spectral properties and molecular brightness of the LSSmScarlet3 LSSRFP (Figure 2b and Table 1). Absorption/excitation/emission maxima for LSSmScarlet3 were observed at 466/466/598 nm, respectively (Figure 2b and Table 1). LSSmScarlet3 had the same large Stokes’ shift of 128 nm as the LSSmScarlet and LSSmScarlet2 proteins. The molecular brightness values at the peak of excitation and at 488 nm for LSSmScarlet3 LSSRFP were 1.34- and 1.31-fold lower than those for LSSmScarlet (Table 1).

We further characterized the photostability of the purified LSSmScarlet2 and LSSmScarlet3 proteins in a one-photon regime using a suspension of microdroplets in oil under continuous irradiation with a metal halide lamp using 470/40 nm light and a 63× oil objective lens (Figure 2e). The developed LSSmScarlet2 and LSSmScarlet3 proteins had similar photobleaching half-times (Table 1). They were 1.5- to 1.6-fold more photostable than LSSmScarlet. In conclusion, both the LSSmScarlet2 and LSSmScarlet3 proteins revealed a one-photon photostability 1.5- to 1.6-fold higher than that of their progenitor LSSmScarlet.

Since the monomeric state of FP ensures its low cytotoxicity and protein tagging in mammalian cells [12,13], we also characterized the oligomeric state of the developed LSSmScarlet2 and LSSmScarlet3 proteins using fast protein liquid chromatography (FPLC) (Figure 2f). LSSmScarlet2 and LSSmScarlet3 LSSRFPs eluted as monomers on FPLC (Figure 2f). Hence, both developed LSSRFPs were monomers and might be suitable for protein labeling.

### 2.6. Chemical Stability of LSSmScarlet2 and LSSmScarlet3 LSSRFPs

As the chemical stability of FPs is important for microscopy methods and for ex vivo imaging, we decided to compare the chemical stability of LSSRFPs under different conditions. We showed above that LSSmScarlet2 and LSSmScarlet3 LSSRFPs demonstrated high pH (Figure 2c), photo- (Figure 2e), and chemical stability in guanidinium chloride (Figure 5b). We next compared the chemical stability of the LSSmScarlet2 and LSSmScarlet3 LSSRFPs in guanidinium chloride incubated for 24 h at room temperature with other monomeric LSSRFPs of different origins, such as mCyRFP1 [14] and mCRISPRed [8] (Figure 6a). We demonstrated that mCyRFP1 and mCRISPRed LSSRFPs were less and more resistant to denaturation with GuaHCl, respectively.

After incubation at room temperature for 24 h in guanidinium thiocyanate (GuaSCN), a stronger denaturant than GuaHCl that has been studied in the context of sfGFP [15], we observed the following rank of stabilities: mCRISPRed > LSSmScarlet2 = mCyRFP1 > LSSmScarlet3 >LSSmScarlet (Figure 6b). Hence, mCRISPRed monomeric LSSRFP withstood guanidinium chloride and guanidinium thiocyanate denaturation better than the other monomeric LSSRFPs tested.

For whole-mount imaging of brain structures, FPs should withstand treatment with 100% methanol [16]. We assessed the LSSRed fluorescence of LSSRFPs after treatment for 1 h at room temperature with 25–100% methanol (Figure 6c). We observed that LSSmScarlet2 and LSSmScarlet3 outperformed other LSSRFPs after 25–100% methanol treatment and, therefore, were the best candidates among LSSRFPs for whole-mount imaging.

For protein-retention expansion microscopy (proExM) [4], the samples were incubated in 1 M NaCl. The fluorescence of some FPs, such as eYFP, was almost totally quenched by 1 M NaCl, but this effect was a consequence of the FPs’ (eYFP’s) chloride sensitivity [17]. We examined LSSRed fluorescence of LSSRFPs after incubation in NaCl solutions for 24 h at room temperature (Figure 6d). All LSSRFPs tested retained their fluorescence after 1 M NaCl treatment and were suitable for proExM.

Endogenous peroxidase activity blocking is one of the most critical steps for successful immunostaining [18]. We showed that the blocking of endogenous peroxidase activity by treatment with 0.3% or 3% H_2_O_2_ at room temperature for 15 min was incompatible with LSSRFP fluorescence because LSSRed fluorescence was totally quenched after such a procedure. Only mCRISPRed LSSRFP retained 50.8% and 8.7% of its fluorescence after treatment with 0.3% and 3% H_2_O_2,_ respectively (Figure 6e).

Cell and tissue fixation is most commonly used in immunohistochemistry and various microscopy techniques. Paraformaldehyde (PFA), glutaraldehyde (GA), and glyoxal (ethanedial) are the most common histological fixatives in biology [19]. If an FP were capable of retaining significant fluorescence after being treated with these fixatives, it would offer significant advantages in optical applications. We coexpressed LSSmScarlet, LSSmScarlet2, or LSSmScarlet3 with EGFP in HeLa mammalian cells and then fixed the cells with 4% PFA or 2.5% GA for 15 min at room temperature (Figure 6f). LSSmScarlet3 retained its fluorescence better than the other LSSRFPs and EGFP after fixation of cells with either PFA or GA. Overall, LSSmScarlet3 outperformed LSSmScarlet in all chemical stability assays and cell fixation; compared to LSSmScarlet2, LSSmScarlet3 had similar chemical stability but retained greater fluorescence after cell fixation.

### 2.7. Brightness of LSSmScarlet2 and LSSmScarlet3 in the Cytosol of Cultured Mammalian Cells

We next compared the brightness of the LSSmScarlet2 and LSSmScarlet3 LSSRFPs with the brightness of LSSmScarlet in the cytosol of mammalian cells. The LSSRFPs were coexpressed with EGFP via the self-cleavable P2A peptide linker in the cytosol of live HeLa cells and visualized under a confocal microscope using a standard 488 nm laser light (Figure 7). Similar to EGFP, both LSSRFPs demonstrated even distribution in the cytosol and nuclei of HeLa cells (Figure 7a,b). The calculated intracellular brightness values for LSSmScarlet2 and LSSmScarlet3 LSSRFPs normalized to the green fluorescence intensity of EGFP were 1.43- and 1.55-fold dimmer than the brightness value for the parental LSSmScarlet (Figure 7c).

### 2.8. Behavior of LSSmScarlet2 and LSSmScarlet3 in Mammalian Cells in Fusions with Structural and Mitochondrial Proteins

The developed LSSmScarlet2 and LSSmScarlet3 LSSRFPs demonstrated monomeric behavior in FPLC. Therefore, we assessed their expression in mammalian cells in fusions with different proteins (Appendix A). When expressed in HeLa cells, LSSmScarlet2 and LSSmScarlet3 demonstrated the correct microtubules, fibers, and intermediate filament-like localization in the fusions with α-tubulin, β-actin, and vimentin cytoskeleton proteins, respectively (Appendix A). Fusions of both LSSRFPs with the dMito signal were properly localized in mitochondrial lumen (Appendix A). The H2B histone protein targeted LSSmScarlet2 and LSSmScarlet3 into the cellular nuclei and did not stop the division of the cells (Appendix A).

### 2.9. Brightness of LSSmScarlet2 and LSSmScarlet3 in Lysosomes of Cultured Mammalian Cells

Since LSSmScarlet2 and LSSmScarlet3 showed extreme pH stability in the acidic pH range up to 2.5, we decided to test their fluorescence in the acidic environment of the lysosomal lumen. The pH of the lysosomal lumen is kept at 4.5–5 [20]. To target LSSmScarlet2, LSSmScarlet3, and the parental LSSmScarlet LSSRFPs in the lysosomal lumen, we fused these proteins with lysosome-associated membrane glycoprotein 3 (LAMP3) [21] and A-isoform of human lysosome associated-membrane protein type 2 (hLAMP2A) [22]. We chose the LAMP3 protein because in a previous study, to develop stable LSSRFP at acidic pH values, the researchers tagged the mCRISPRed protein into the lysosomal lumen using LAMP3 [8]. hLAMP2A was chosen since hLAMP2A is a marker of chaperone-mediated autophagy (CMA), a specific lysosome-dependent protein degradation pathway [23]. During CMA, the complex of chaperone HSC70 with the substrate protein tagged for degradation interacts with the cytosolic C-terminal tail of LAMP-2A that, in turn, drives the translocation of the target protein into the lysosome lumen [24]. Reduced expression of the lamp2 gene has been related to genetic polymorphisms [25], aging [26], and metabolic disorders [27]. Hence, the search for pH-stable FPs suitable for labeling the *N*-terminus of the hLAMP2A protein will help visualize CMA autophagy.

We coexpressed LSSRFP-LAMP3 and LSSRFP-hLAMP2A fusions with EGFP in HeLa mammalian cells. To take into account the influence of the LSSRFP expression level on brightness, we normalized LSSRed fluorescence to green fluorescence. For LAMP3 fusions, we observed mainly lysosomal punctate localization for all LSSRFPs (Figure 8). The brightness of LSSmScarlet3 in the LAMP3 fusion surpassed by 2.16-fold the brightness of LSSmScarlet, whereas the brightness of LSSmScarlet2 was similar to that of the parental LSSmScarlet (Figure 8b). Therefore, the higher pH stability of LSSmScarlet3 together with its fast maturation favored its higher brightness inside the lysosome lumen. For hLAMP2A fusions, we noted that LSSmScarlet and LSSmScarlet3 had mainly punctate localization characteristic of lysosomes, and LSSmScarlet2 had mainly cytosolic localization and rarely punctate localization in cells with low expression (Figure 9). Estimation of the brightness in hLAMP2A fusions showed that LSSmScarlet3 had approximately the same brightness as LSSmScarlet, but LSSmScarlet2 was 2.25-fold dimmer than LSSmScarlet (Figure 9b). Hence, LSSmScarlet3 had proper localization in fusion with LAMP3 and hLAMP2A proteins and compared to LSSmScarlet had superior and the same brightness in fusion with LAMP3 and hLAMP2A proteins, respectively.

## 3. Materials and Methods

### 3.1. Cloning of Bacterial Vectors, Mutagenesis, and Library Screening

Cloning of bacterial plasmids, mutagenesis, and library screening were performed as described earlier [9] using primers listed in Appendix A. For details, please see the Appendix A.

### 3.2. Protein Purification and Characterization

Purification and characterization of LSSRFPs and preparative purification of LSSmScarlet2 protein for X-ray crystallography were performed as described earlier [9]. The protein was concentrated to 10 mg/mL in 20 mM Tris-Cl, 0,2 M NaCl, and pH 7.8 buffer. The raw photobleaching data are shown in Appendix A.

### 3.3. Protein Crystallization

LSSmScarlet2 was crystallized by the sitting drop vapor diffusion method using 48-well Intelli plates (Art Robbins) at 20 °C. The crystallization conditions from [9] were used as initial conditions. Plate-like crystals were grown within one week in the following conditions: 0.1 M lithium sulfate, 0.1 M MES pH 6.0, and 25% PEG 3350.

### 3.4. Data Collection, Processing, Structure Solution, and Refinement

LSSmScarlet2 crystals were briefly soaked in artificial mother liquor containing 20% glycerol (*v*/*v*) as a cryoprotectant and flash-frozen in liquid nitrogen. The X-ray data were collected from a single crystal at 100 K to a resolution of 1.41 Å at beamline ID30A of the ESRF, Grenoble, France. The data were indexed and integrated using the package XDS [28] and scaled using the Aimless program of the CCP4 suite [29]. The experimental details are summarized in Table 2. LSSmScarlet2 was obtained in space group C2. The asymmetric unit contains one molecule.

The X-ray structure of the LSSmScarlet2 protein was solved by molecular replacement with the MOLREP program [30]. The structure of the red fluorescent protein mScarlet (PDB ID 5LK4) was used as an initial model. Model building and refinement were carried out using COOT [31] and REFMAC5 [29], respectively. The atomic coordinates and experimental data have been deposited in the Protein Data Bank under ID code 8ARM.

### 3.5. Structure Analysis and Validation

Visual inspection and analysis of the structure were carried out using the COOT program and the PyMOL Molecular Graphics System, Version 1.9.0.0 (Schrödinger, San Francisco, CA, USA). The interface between molecules in the crystal was analyzed using PDBePISA [11].

### 3.6. Chemical Stability

To study the chemical stability of the purified proteins in guanidinium chloride, we added 2 μL of 10 μM LSSRFPs to 200 μL of 30 mM HEPES (pH 7.6) buffer supplemented with 0, 0.5, 1, 1.5, 2, 2.5, 3, 3.5, 4, 4.5, 5, 5.5, 6, or 6.3 M guanidinium chloride and incubated the solutions at room temperature for 24 h. Then, the LSS red fluorescence (Ex 490 nm/Em 580–640 nm) was measured using a 96-well ModulusTM II Microplate Reader (Turner Biosystems, Sunnyvale, CA, USA).

To study the chemical stability of purified proteins in guanidinium thiocyanate, we added 2 μL of 10 μM LSSRFPs to 200 μL of 30 mM HEPES (pH 7.6) buffer supplemented with 0, 0.5, 1, 1.5, 2, 2.5, 3, or 3.6 M guanidinium thiocyanate and incubated the mixtures at room temperature for 24 h. Then, the LSS red fluorescence was measured.

To study the chemical stability of purified proteins in the H_2_O_2_ solution, we added 2 μL of 10 μM LSSRFPs to 200 μL of PBS (pH 7.4) buffer supplemented with 0, 0.3, or 3% H_2_O_2_ and incubated the mixtures at room temperature for 15 min. Then, LSS red fluorescence was measured.

To study the chemical stability of purified proteins in methanol, we added 2 μL of 10 μM LSSRFPs to 200 μL of PBS (pH 7.4) buffer supplemented with 0, 25, 50, 75, or 100% methanol and incubated the mixtures at room temperature for 1 h. Then, LSS red fluorescence was measured.

To study the chemical stability of the purified proteins in sodium chloride solution, we added 2 μL of 10 μM LSSRFPs to 200 μL of 30 mM HEPES (pH 7.6) buffer supplemented with 0, 50, 100, 200, 400, 500, or 1000 mM sodium chloride and incubated the mixtures at room temperature for 24 h. Then, the LSS red fluorescence was registered.

### 3.7. Mammalian Plasmid Construction

Mammalian plasmid construction was performed as described in [9,32,33]. For details, please see the Appendix A.

### 3.8. Mammalian Live- and Fixed-Cell Imaging

Cultivation, transient transfection of HeLa Kyoto cells and imaging were performed as described earlier [9].

### 3.9. Cell Fixation with 4% PFA and 2.5% GA

Cells were chemically fixed as described earlier [9]. For details, please see the Appendix A.

### 3.10. Statistics

To estimate the significance of the difference between two values, we used the Mann–Whitney Rank Sum Test and determined the *p* values (throughout the text in the brackets) for the two-tailed hypothesis. We considered the difference significant if the *p* value was <0.05.

## 4. Conclusions

In conclusion, we developed and characterized two pH-, photo-, and chemically stable LSSRFPs, named LSSmScarlet2 and LSSmScarlet3. Both proteins demonstrated a lower brightness at 488 nm excitation than the parental LSSmScarlet, both as purified proteins (Table 1) and in the cytosol of mammalian cells (Figure 7). However, they maintained 100% fluorescence in the pH range from 2.5 to 10 and had pKa values of 2.2, in contrast to the parental LSSmScarlet, whose fluorescence was characterized by two pKa values of 1.91 and 5.78 [9]. Although the mCRISPRed and CRISPRed2 LSSRFPs had one pH transition at acidic pH with pKa values of 2.14–2.23, they were characterized by an approximately 2-fold decrease in their fluorescence at pH values from 7 to 10 with pKa values of approximately 10 [9]. Therefore, LSSmScarlet2 and LSSmScarlet3 LSSRFPs together with mTagBFP [34] and mTagBFP2 [7] blue FPs are the most pH stable among all FPs known today.

LSSmScarlet2 and LSSmScarlet3 demonstrated 1.5- to 1.6-fold higher photostability (Figure 2e) and higher chemical stability to resist denaturation by guanidinium chloride, guanidinium thiocyanate, and methanol (Figure 6a–c) than the progenitor LSSmScarlet. Their better stability in methanol makes the LSSmScarlet2 and LSSmScarlet3 proteins the best candidates among LSSRFPs for whole-mount imaging of brain structures.

Additionally, we solved the X-ray structure of the LSSmScarlet2 protein, performed site-directed mutagenesis, and found the key role of the T74 residue in the pH and chemical stability of the LSSmScarlet2 and LSSmScarlet3 proteins.

LSSmScarlet3 showed the fastest maturation among LSSRFPs of the LSSmScarlet series (Table 1): its maturation half-time was 1.8- and 5.4-fold shorter than the half-times of LSSmScarlet and LSSmScarlet2, respectively. When expressed in mammalian cells, LSSmScarlet3 preserved the greatest LSSRed fluorescence after fixation of cells with paraformaldehyde or glutaraldehyde solutions (Figure 6f).

Finally, because of its high pH stability together with its fastest maturation, LSSmScarlet3 demonstrated the highest brightness among LSSRFPs of the LSSmScarlet series in the lysosomal lumen when expressed in LAMP3 fusions. It was 2.2- and 2.6-fold brighter than LSSmScarlet and LSSmScarlet2, respectively. When expressed inside the lysosomal lumen in hLAMP2A fusions, the brightness of LSSmScarlet3 was similar to that of LSSmScarlet.

## Figures and Tables

**Figure 1 ijms-23-11051-f001:**
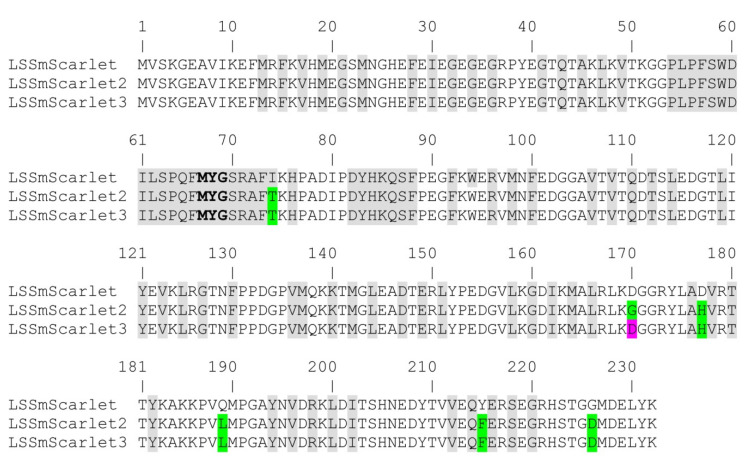
Alignment of the amino acid sequences for the developed LSSRFPs (LSSmScarlet2 and LSSmScarlet3) and the parental protein (LSSmScarlet). Mutations in the developed LSSRFPs are highlighted in green. The mutation in LSSmScarlet3 compared to LSSmScarlet2 is in magenta. Chromophore-forming residues MYG are selected in bold. Residues buried in the β-barrel are in gray.

**Figure 2 ijms-23-11051-f002:**
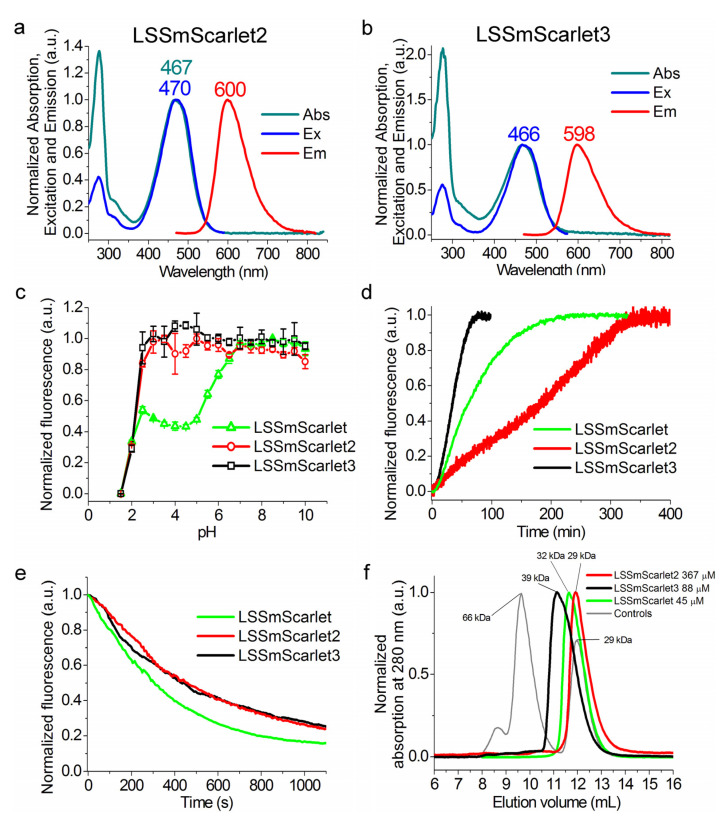
In vitro properties of the purified LSSRFPs. (**a**,**b**) Absorption, excitation, and emission spectra for the LSSRFPs in PBS buffer at pH 7.40. (**c**) Red fluorescence intensity for LSSRFPs as a function of pH. (**d**) Maturation of LSSRFPs in PBS buffer at pH 7.40. (**e**) Photobleaching of LSSRFP droplets in oil under continuous wide-field imaging using a metal halide lamp (3.78 mW/cm^2^ light power before the objective lens). (**c**–**e**) Three to six replicates were averaged for analysis. The error bars represent the standard deviations. (**f**) Fast protein liquid chromatography of LSSRFPs. LSSRFPs were eluted in 20 mM Tris-HCl (pH 7.80) and 200 mM NaCl buffer. The molecular weights of LSSRFPs were calculated from a linear regression of the dependence of the logarithm of the control molecular weights vs. elution volume (Appendix A).

**Figure 3 ijms-23-11051-f003:**
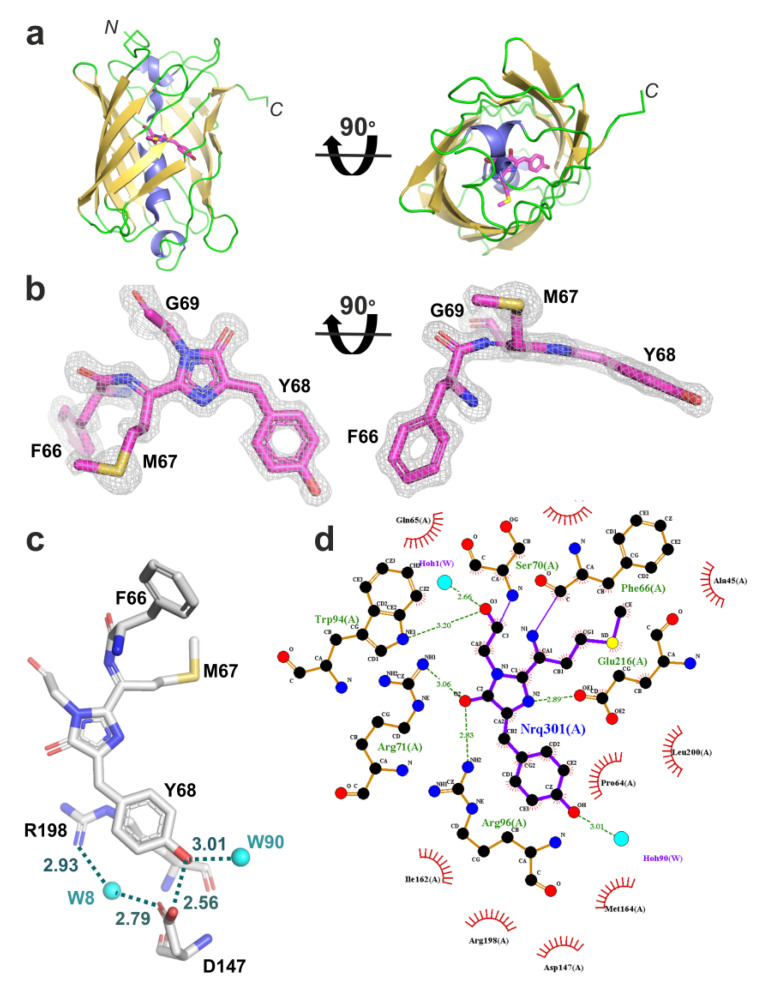
X-ray structure of the LSSmScarlet2 protein. (**a**) Cartoon representation of the overall LSSmScarlet structure. β-Sheets, α-helixes, and unstructured regions are shown in yellow, blue, and green, respectively. Chromophore carbon atoms are colored in magenta. The orientation of the panel on the right is rotated 90° around the horizontal axis with respect to that on the left. (**b**) 2Fobs-Fcalc electron density map is shown for the chromophore of the LSSmScarlet2 protein. The electron density map is shown as a gray mesh and contoured at the 1.5σ level. The orientation of the chromophore on the right is rotated 90° around the horizontal axis with respect to that on the left. (**c**) Hydrogen bond network around the phenolic hydroxyl group of the LSSmScarlet2 chromophore. (**d**) Immediate environment of the LSSmScarlet2 chromophore.

**Figure 4 ijms-23-11051-f004:**
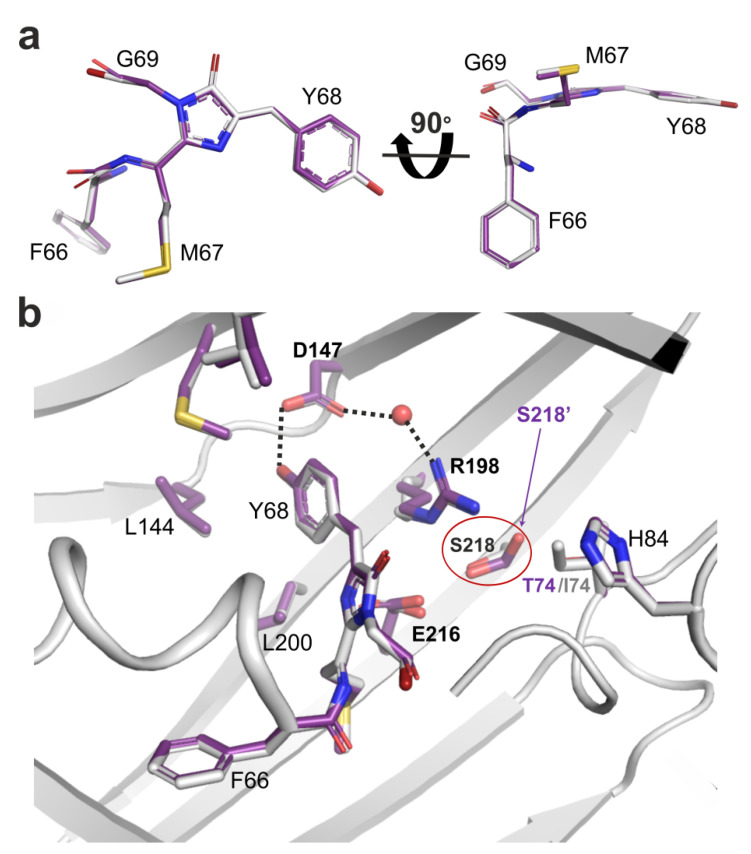
Structural comparison of the chromophores (**a**) and their immediate environments (**b**) for the LSSmScarlet2 (carbon atoms are colored magenta) and LSSmScarlet (PDB: 7OIN) (carbon atoms are colored gray) proteins. A water molecule (red sphere) provides hydrogen bonding (dashed lines) between residues D147 and R198 of LSSmScarlet2. D147 of LSSmScarlet2 forms a direct hydrogen bond (dashed line) with the Y68 residue. S218 in LSSmScarlet2 is highlighted by a red oval, and the alternative conformation of S218 (S218′) is indicated by an arrow. In Panel (**a**), the orientation of the chromophore on the right is rotated 90° around the horizontal axis with respect to that on the left.

**Figure 5 ijms-23-11051-f005:**
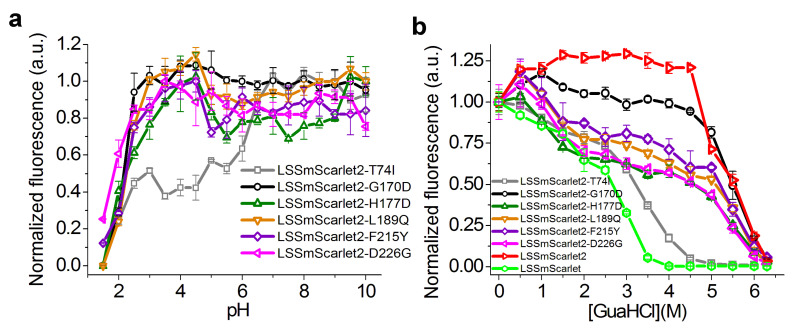
pH stability (**a**) and chemical stability in guanidinium chloride (**b**) of LSSmScarlet2 mutants.

**Figure 6 ijms-23-11051-f006:**
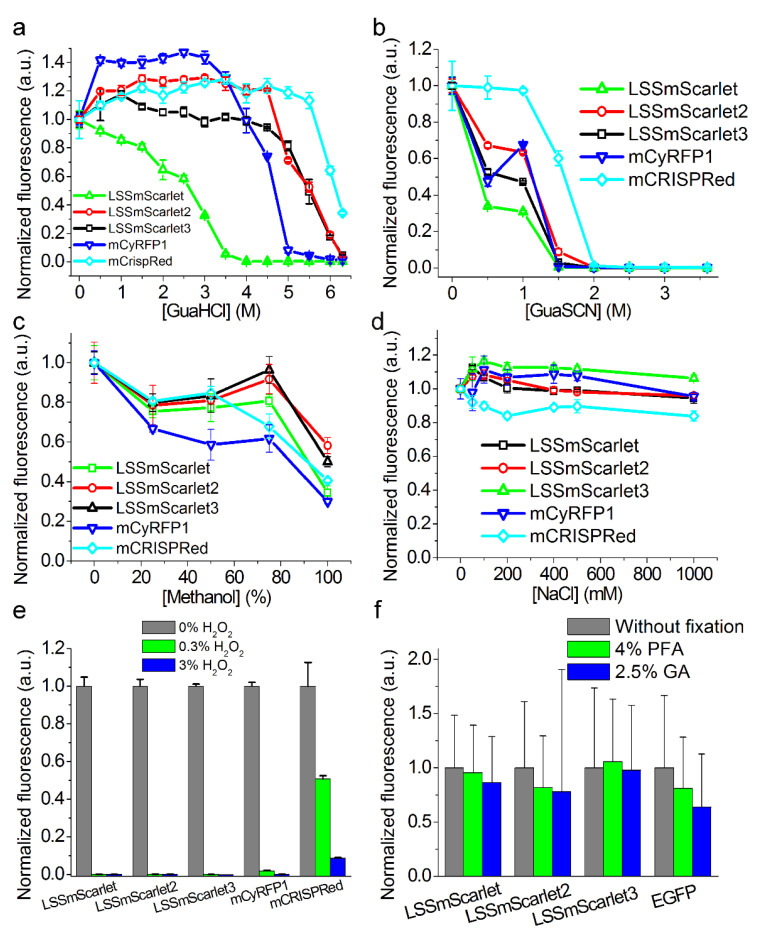
Chemical stability of LSSmScarlet, LSSmScarlet2, and LSSmScarlet3. Purified proteins (0.1 μM) were incubated (**a**) for 24 h at room temperature in 0, 0.5, 1, 1.5, 2, 2.5, 3, 3.5, 4, 4.5, 5, 5.5, 6, and 6.3 M GuaHCl; (**b**) for 24 h at room temperature in 0, 0.5, 1, 1.5, 2, 2.5, 3, and 3.6 M GuaSCN; (**c**) for 1 h at room temperature in 0, 25, 50, 75, and 100% methanol; (**d**) for 24 h at room temperature in 0, 50, 100, 200, 400, 500, and 1000 mM NaCl; or (**e**) for 15 min at room temperature in 0, 0.3, or 3% H_2_O_2_. (**f**) LSSmScarlet, LSSmScarlet2, or LSSmScarlet3 was coexpressed with EGFP in HeLa Kyoto mammalian cells, and then the cells were washed with DPBS and fixed with 4% PFA or 2.5% GA at room temperature for 15 min.

**Figure 7 ijms-23-11051-f007:**
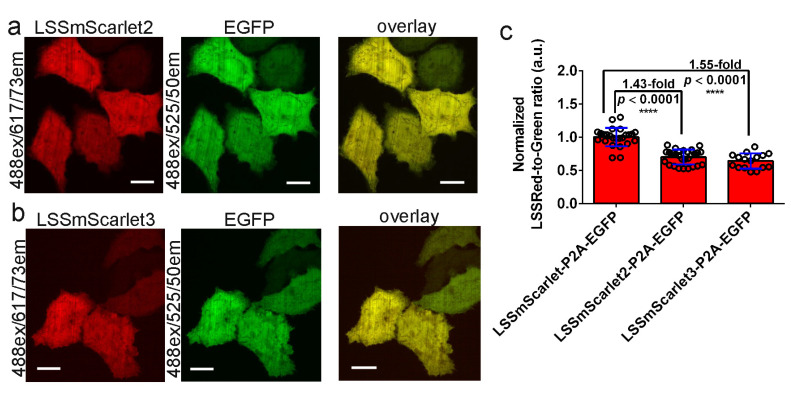
Comparison of the brightness of the LSSRFPs in the cytosol of HeLa cells. (**a**,**b**) Confocal images of HeLa cells expressing the LSSmScarlet2-P2A-EGFP (**a**) or LSSmScarlet3-P2A-EGFP (**b**) proteins. LSSRed (488 ex and 617/73 em, in red color) and green (488 ex and 525/50 em, in green color) fluorescence channels and their overlay are shown. Scale bar, 50 µm. (**c**) The averaged brightness for the LSSRFPs in HeLa cells normalized to the brightness of the EGFP expressed in the same cells. The EGFP is connected to LSSRFPs via a P2A-self cleavable linker. The error bars are the standard deviations across sixteen to twenty-eight cells. The *p* values show significant differences between the respective values. ****, *p* value is <0.0001.

**Figure 8 ijms-23-11051-f008:**
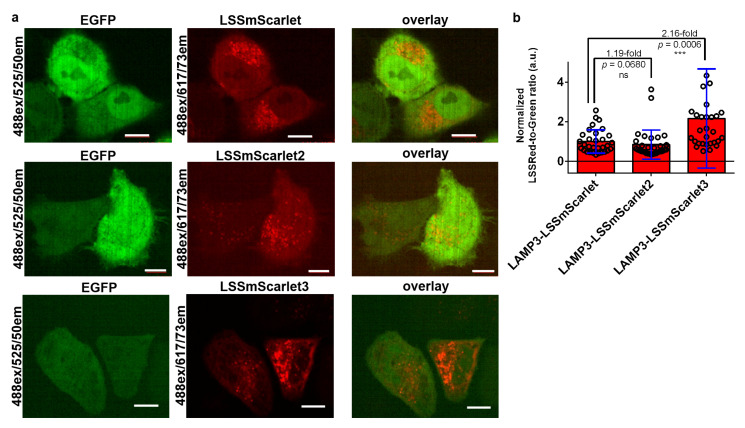
Comparison of the brightness of the LSSRFPs in fusions with LAMP3 inside the lysosomal lumen in HeLa cells. (**a**) Confocal images of HeLa cells coexpressing EGFP with the LAMP3-LSSmScarlet, LAMP3-LSSmScarlet2, or LAMP3-LSSmScarlet3 LSSRFP. LSSRed (488 ex and 617/73 em, in red color) and green (488 ex and 525/50 em, in green color) fluorescence channels and their overlay are shown. Scale bar, 50 µm. (**b**) Averaged brightness for the LSSRFPs in lysosomes of HeLa cells normalized to the brightness of the EGFP expressed in the same cells. The error bars are the standard deviations across 29–32 cells. The *p* values show significant differences between the respective values. ns, not significant, *p* value > 0.05. ***, *p* value < 0.001.

**Figure 9 ijms-23-11051-f009:**
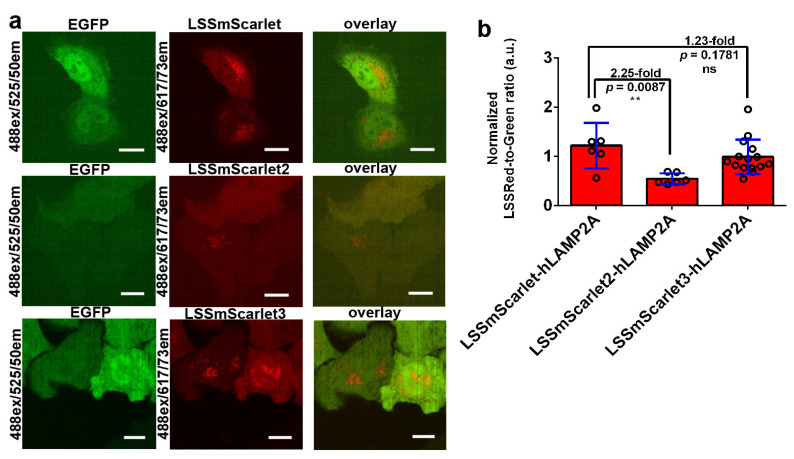
Comparison of the brightness of the LSSRFPs in fusions with hLAMP2A inside the lysosomal lumen of HeLa cells. (**a**) Confocal images of HeLa cells coexpressing EGFP with the LSSmScarlet-hLAMP2A, LSSmScarlet2-hLAMP2A, or LSSmScarlet3-hLAMP2A LSSRFPs. LSSRed (488 ex and 617/73 em, in red color) and green (488 ex and 525/50 em, in green color) fluorescence channels and their overlay are shown. Scale bar, 50 µm. (**b**) The averaged brightness for the LSSRFPs in lysosomes of HeLa cells normalized to the brightness of the EGFP expressed in cytosol of the same cells. The error bars are standard deviations of 6–15 cells. The *p* values show the significant differences between the respective values. ns, not significant, *p* value > 0.05. **, *p* value < 0.01.

**Table 1 ijms-23-11051-t001:** In vitro properties of the LSSRFPs. ^a^ QY was determined at pH 7.40. LSSmScarlet (QY = 0.43 [9]) was used as the reference standard. ^b^ Extinction coefficients were determined by the alkaline denaturation method. ^c^ Molecular brightness was calculated as a product of the quantum yield and extinction coefficient at maximal absorption and normalized to the brightness of EGFP, which has an extinction coefficient of 56,000 M^−1^·cm^−1^ and quantum yield of 0.6 [1]. ^d^ Brightness corrected for the decreased absorption at 488 nm. ^e^ Half-time to bleach until 50%. One-photon photobleaching was performed under a metal halide lamp on droplets of protein solution (45 µM final protein concentration) in oil. The standard deviations are shown.

Protein	Abs, Ex/Em (nm)	QY (%) ^a^	ε (mM^−1^·cm^−1^) ^b^	Brightness vs. EGFP (%)		pKa	Monomeric State	Photobleaching Half-Time (s) ^e^	Maturation Half-Time (min)
				Ex. at Max ^c^	Ex. at 488 nm ^d^				
**LSSmScarlet**	466, 470/598	43 ± 2	30.2 ± 0.6	39	34	1.91 ± 0.01; 5.78 ± 0.06	Monomer	310 ± 53	61
**LSSmScarlet2**	467, 470/600	29 ± 2	30.0 ± 1.2	26	23.5	2.19 ± 0.01	Monomer	487 ± 148	184
**LSSmScarlet3**	466,466/598	36 ± 1	27.3 ± 1.4	29	26	2.18 ± 0.01	Monomer	463 ± 237	34

**Table 2 ijms-23-11051-t002:** Data collection and refinement.

Data Collection	
Diffraction Source	ESRF, ID30A
Wavelength (Å)	0.967697
Resolution range (Å)	39.8–1.41
Detector	Eiger 4M
Space group	C2
*a*, *b*, *c* (Å)	84.62; 45.37; 58.74
α, β, γ (°)	90.0; 102.09; 90.0
Unique reflections	42,307 (2725)
Resolution range (Å)	39.8–1.41 (1.45–1.41) *
Completeness (%)	97.57 (91.7)
Multiplicity	1.56 (1.38)
〈*I*/σ(*I*)〉	15.5 (4.43)
R_meas_ (%)	5.8 (25.8)
CC_1/2_ Wilson B-factor (Å^2^)	99.8 (93.8) 17.61
**Refinement**	
Resolution range	15–1.41
*R*_work_/*R*_free_ (%)	11.25/15.23
Bonds (Å)	0.011
Angles (°)	1.7525
Ramachandran plot	
Most favored (%)	95.5
Allowed (%)	4.5
No. atoms	
Protein	1888
Water	381
Chromophore	23
Other ligands	5
B-factors (Å^2^)	
Protein	10.8
Water	21.34
Chromophore	9.7
Other ligands	21.8

* Values in parentheses are for the highest-resolution shell.

**Table 3 ijms-23-11051-t003:** pH stability and chemical stability of LSSmScarlet, LSSmScarlet2, LSSmScarlet3 (LSSmScarlet2/G170D), and LSSmScarlet2 mutants in guanidinium chloride.

Protein	pKa	GuaHCl Concentration at Half-Maximal Fluorescence (M)
**LSSmScarlet**	**1.91 ± 0.01;** **5.78 ± 0.06**	2.66 ± 0.10
**LSSmScarlet2**	2.19 ± 0.01	5.50 ± 0.05
**LSSmScarlet3**	2.18 ± 0.01	5.49 ± 0.13
**LSSmScarlet2/T74I**	2.03 ± 0.04 6.16 ± 0.09	3.20 ± 0.06
**LSSmScarlet2/H177D**	2.22 ± 0.16	4.55 ± 0.13
**LSSmScarlet2/L189Q**	2.33 ± 0.02	4.86 ± 0.44
**LSSmScarlet2/F215Y**	2.25 ± 0.01	5.25 ± 0.09
**LSSmScarlet2/D226G**	1.88 ± 0.06	4.61 ± 0.36

## Data Availability

The data are contained within the article or its Appendix A.

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
