# Peer review of "LSSmScarlet2 and LSSmScarlet3, Chemically Stable Genetically Encoded Red Fluorescent Proteins with a Large Stokes’ Shift"

_ijms, 2022, doi:10.3390/ijms231911051_

Round 1

Reviewer 1 Report

The authors describe two new versions of LSSmScarlet. They analyze the their properties in detail and from several directions. They find that these two new versions, called LSSmScarlet2 and LSSmScarlet3, are chemically stable and stable in different pH environments. Such probes, especially in combination with the large Stokes shift, are extremely useful for many applications and interesting for a wide audience. The manuscript is in very good shape and can be published as is. Congratulations to the authors - great work!

I would recommend only one small modification:

In Fig 1, the the green and magenta background is to strong and therefore the sequence can barely be read. Please make it more transparent.

Reviewer 2 Report

This manuscript can be after a revision. The following concerns must be answered.

(1). The correct abbreviation is "Stokes' shift", this should be corrected throughout the manuscript.

(2). In the abstract it will be better to highlight the Stokes' shift improvements.

(3). For the photobleaching study, it is important to provide physical parameters such as the power/efficiency of the lamp and pulse time intervals. Also actual emission plots must be provided.

(4). Authors should highlight clearly, how does the improved Stokes' shift would help in experimental settings.

(5). There are few grammatical and textual errors that should be answered.

Round 2

Reviewer 2 Report

Accept in the present form